# Multiple Genetic Loci Associated with Pug Dog Thoracolumbar Myelopathy

**DOI:** 10.3390/genes14020385

**Published:** 2023-02-01

**Authors:** Gustaf Brander, Cecilia Rohdin, Matteo Bianchi, Kerstin Bergvall, Göran Andersson, Ingrid Ljungvall, Karin Hultin Jäderlund, Jens Häggström, Åke Hedhammar, Kerstin Lindblad-Toh, Katarina Tengvall

**Affiliations:** 1Science for Life Laboratory, Department of Medical Biochemistry and Microbiology, Uppsala University, 751 23 Uppsala, Sweden; 2Broad Institute of MIT and Harvard, Cambridge, MA 02142, USA; 3Department of Clinical Sciences, Swedish University of Agricultural Sciences, 750 07 Uppsala, Sweden; 4Department of Animal Breeding and Genetics, Swedish University of Agricultural Sciences, 750 07 Uppsala, Sweden; 5Department of Companion Animal Clinical Sciences, Norwegian University of Life Sciences, N-1432 Ås, Norway

**Keywords:** pug, myelopathy, GWAS, dog genetics, BayesR, bayesian, selection, XP-EHH, bone homeostasis, cartilage, inflammatory response, fibrotic scar tissue

## Abstract

Pug dogs with thoracolumbar myelopathy (PDM) present with a specific clinical phenotype that includes progressive pelvic limb ataxia and paresis, commonly accompanied by incontinence. Vertebral column malformations and lesions, excessive scar tissue of the meninges, and central nervous system inflammation have been described. PDM has a late onset and affects more male than female dogs. The breed-specific presentation of the disorder suggests that genetic risk factors are involved in the disease development. To perform a genome-wide search for PDM-associated loci, we applied a Bayesian model adapted for mapping complex traits (BayesR) and a cross-population extended haplotype homozygosity test (XP-EHH) in 51 affected and 38 control pugs. Nineteen associated loci (harboring 67 genes in total, including 34 potential candidate genes) and three candidate regions under selection (with four genes within or next to the signal) were identified. The multiple candidate genes identified have implicated functions in bone homeostasis, fibrotic scar tissue, inflammatory responses, or the formation, regulation, and differentiation of cartilage, suggesting the potential relevance of these processes to the pathogenesis of PDM.

## 1. Introduction

The pug has been bred to promote characteristics such as small stature, flat face, large head, bulging eyes, wrinkled forehead, and soft, thin ears. The selection for specific traits and a small population size has contributed to a strong predisposition to several disorders. The 2020 Breed Health and Conservation Plan for Pugs included spinal problems in their list of priority health issues [1], with a two-fold increase in the risk of spinal cord disorder in pugs compared to other breeds [2]. The orthopedic foundation of America has reported that pelvic limb abnormalities are the second most common health concern in the pug breed. In addition, the insurance data from the biggest animal insurance company in Sweden, Agria Pet Insurance, has shown a seven-fold increase in the mortality rate in pugs with ataxia and paresis, which are the clinical consequences of a spinal cord problem. Pug dogs with thoracolumbar myelopathy (PDM) commonly present with a chronic, progressive clinical course of ataxia and paraparesis, often accompanied by incontinence [3,4,5,6,7]. Although no exact prevalence of PDM has been established, gait abnormalities (including, but not exclusively, PDM) are a common problem in pugs [4].

PDM has a late age of onset and usually exhibits neurological deficits at the age of around seven [5,7,8]. In Sweden, the affected pugs are usually euthanized within a year of the onset of pelvic limb ataxia [8]. The histopathologic characteristics of PDM include meningeal fibrosis with associated spinal cord destruction, often accompanied by neighboring vertebral column pathology. In addition, central nervous system (CNS) inflammation has been reported in a considerable number of pugs with PDM [8]. Differences in meningeal and vertebral column pathologies have led to various proposed etiologies, however shared pathological findings suggest they belong to a spectrum of the same disorder. Due to the breed-specific appearance of PDM, there are likely genetic risk factors contributing to the condition, although no genes associated with the disorder have hitherto been reported.

In this study, we analyzed 89 pugs (51 cases and 38 controls) with the aim of identifying the genes associated with PDM. First, we applied a Bayesian model adapted to complex trait mapping, BayesR, a well-suited method when defining multiple genetic risk factors with smaller effects, which is common for complex traits. BayesR assesses the effect size variances of all genetic variants simultaneously, resulting in a lower risk of false negatives. Moreover, by treating the effect sizes as random effects, the corresponding estimates are unbiased [9]. Second, we performed a cross-population extended haplotype homozygosity analysis (XP-EHH), which aims to detect candidate regions under recent selection by comparing the integrated extended haplotype homozygosity (EHH) in two subpopulations [10,11]. This methodology was employed to explore the possibility that the increased risk for PDM derives from pleiotropic or hitchhiking effects resulting from artificial selection. The selection of the specific pug characteristics could have created large genetic regions with high linkage disequilibrium (LD) and, if it is associated with PDM risk, this could be detected as an extended haplotype homozygosity differing between cases and controls. In total, we identified 19 associated loci with BayesR and three candidate regions under selection. These loci and regions harbor candidate genes with implicated functions in bone homeostasis, cartilage, fibrotic scar tissue, and inflammatory responses, and are therefore potentially relevant to the pathogenesis of PDM.

## 2. Materials and Methods

### 2.1. Ethical Statement

The study was approved by the Local Ethical Committees in Uppsala and Stockholm, Sweden. Signed informed consent was obtained from all of the dog owners.

### 2.2. Study Population

All of the dogs included in this study were privately owned and sampled in collaboration with veterinary clinics in Sweden, between 2010–2020. In addition, samples were included from phenotyped dogs that had donated blood to the biobank at the Swedish University of Agricultural Sciences, Uppsala, Sweden.

The study was comprised of affected pugs with signs of a myelopathy localized to the thoracolumbar spinal cord and control pugs without these signs. The affected pugs presented with a history of more than one month of a bilateral pelvic limb gait abnormality, described by the owner as incoordination and weakness. In addition, the author, C. Rohdin, performed a neurological examination of potential cases to confirm pelvic limb ataxia and paraparesis, suggestive of a myelopathy localized to the thoracolumbar spinal cord. Magnetic resonance imaging was performed, as previously described in Rohdin et al. [8], to confirm the focal spinal cord pathology in 37 affected pugs. We have previously shown that pugs presenting with a history of more than one month of bilateral pelvic limb gait abnormality alongside a neurological examination confirming pelvic limb ataxia and paraparesis, all had spinal cord pathology involving the meninges and spinal cord [8]. A subset of the recruited dogs was also included in two previous studies investigating the presence of vertebral malformations and pathology, respectively, of pugs with PDM [4,8].

The control pugs were at least eight years old with no history or signs of a neurological disorder. Specifically, to be included as a control, no signs of incoordination, weakness or incontinence should be present; nor should they exhibit worn-down nails or skin on the dorsum of the paws in the pelvic limbs.

Before quality control, the affected pugs consisted of 27 females (40.3%) with an average age at onset of 81.5 months (SD 23.7, median 84, interquartile range 68–96.5), and the control pugs consisted of 28 females (60.9%) with an average age at phenotyping of 132 months (SD 18.9, median 129, interquartile range 116–145).

Blood samples were collected in EDTA tubes and stored in −70 °C until analysis. One subset of the sample set, which included 31 pugs (10 cases/21 controls), was genotyped using Illumina 230k CanineHD BeadChip (Illumina, San Diego, CA, USA), and the other subset, 86 pugs (60 cases/26 controls), was whole-genome sequenced at low coverage using Gencove’s low-pass sequencing followed by imputation against a reference panel of 676 dogs (Pipeline Dog low pass v2; Gencove Inc, New York, NY, USA). The chromosomal coordinates were based on the dog CanFam3.1 genome assembly [12].

### 2.3. Imputation

The Illumina genotyped pugs underwent genetic imputation using the 86 Gencove low-pass sequenced pugs as the reference panel. To remove low confidence genotypes, variants that were not present in the reference panel were excluded prior to imputation, as were variants with minor allele frequency (MAF) < 0.05 and call rate < 95%. One case dog from the Illumina dataset with a <95% call rate was excluded. The remaining 30 dogs were phased using Shapeit v4.1.3 [13] and the imputation was performed using Impute2 v2.3.2 [14]. To validate the imputation, a selection of 5% random variants (*n* = 4965) were removed from the Illumina genotype dataset before imputing a second time. For this set of variants, the concordance rate between the imputed genotypes and the Illumina genotypes was 96.7%. 

### 2.4. Quality Control

Quality control (QC) was performed using PLINK v1.9. [15]. After imputation, all individuals and variants had a call rate of >95%. The Hardy-Weinberg Equilibrium threshold, set to 1e−10 for both the cases and controls, removed 475 variants, and 24,745 variants with MAF below 0.01 were removed. The multidimensional scaling revealed no clusters or outliers. To correct for any batch effect deriving from the different genotyping technologies used in the two sample sets, we contrasted these two groups and found 6229 variants with a statistical difference at *p* < 1e−03, which we subsequently discarded. After QC, 2,140,239 variants remained. All of the dogs showed concordance between the reported and genotype predicted sex, and no dogs had an outlying (3 SD) average heterozygosity rate.

Using KING v2.2.7 to test for relatedness [16], five duplicates were identified within or between the genotyping technique groups. The duplicates were confirmed from the official records of individual registration numbers. An additional 22 pairs of pugs with a relatedness score above 0.177 (corresponding to a kinship between first- and second-degree relatives) were identified. From the duplicate pairs and the pairs above the relatedness threshold of 0.177, the individual with the lower call rate was excluded, resulting in 89 pugs (51 cases and 38 controls) in the final dataset.

### 2.5. Genetic Analyses

We used R v4.0.2 [17] and the following R-packages: GWASTools v1.40.0 [18] and GENESIS v2.24.0 [19] for handling the genotype data, and SNPRelate v1.28.0 [20] for defining the PCs and kinship. All of the figures were generated with ggplot2 v.3.3.5 [21].

The BayesR software (v. 1, updated 1 April 2021) implements a Bayesian mixture model for the analysis of complex traits, assuming that the variant effects belong to four different variance distributions: zero effect, 0.01σ^2^ g, 0.001σ^2^ g, and 0.0001σ^2^ g. Markov Chain Monte Carlo sampling is applied to arrive at the posterior inference about the variant effects based on the four distributions. Random effects, rather than fixed effects, provide less biased estimates than the traditional GWAS models and, as the effect of all variants are assessed simultaneously, the analysis is not subjected to multiple testing [9]. Before running the BayesR model, we LD-pruned the imputed QCed dataset using Plink v1.9 (window size 25 kb, step size 5 kb, correlation threshold [r^2^] 0.999). The Illumina genotyped variants that had been deleted in the LD-pruning were then merged back in. The final dataset consisted of 266,687 variants. The model, adjusted for genotyping technique (Illumina vs. Gencove) and sex, was run using 300,000 iterations and 100,000 burn-in samples, with every tenth sample saved for posterior inference. This was repeated five times to assess the convergence of the model, and the absolute values of the average variant effects were reported as the final results. The top 50 effect size variants were defined as effect variants and the associated loci were defined by variants in LD at r^2^ > 0.8. The sum of the risk alleles at the top effect variant per associated locus (risk index) was compared between cases and controls using the two-tailed Welsh Two Sample T test, and the phenotypic variance explained by the associated loci and sex, was calculated using ANOVA (R package stats v. 4.1.2).

To investigate whether the artificial selection for desired pug characteristics has resulted in the accumulation of risk or protective factors for PDM, we performed a selection signature analysis, comparing cases and controls. First, we used fastPHASE v.1.4.8 [22] with random starts set to 10, to identify the haplotypes for cases and controls and then the R package rehh v3.2.1 [23,24] to identify EHH in cases and controls separately. EHH detects the transmission of an extended haplotype without recombination [10] and XP-EHH compares the integrated EHH between two populations at each variant position [11]. XP-EHH detects regions with an extended haplotype around the alleles at a position in one population, but with a more rapid decay of LD in the other. We compared cases to the controls; thus, the positive XP-EHH scores indicate the selection in cases, whereas the negative scores indicate the selection in controls. The candidate regions under selection were defined by the variants with −log10(p) XP-EHH > 5.

### 2.6. Data Availability

The genotype datasets (plink files: bed, bim, fam, pheno), after quality control and relatedness filtering, have been uploaded onto the SciLifeLab data repository (DOI 10.17044/scilifelab.21948521).

## 3. Results

The final sample set consisted of 89 pugs (51 PDM cases and 38 controls). Thirty-two of the affected pugs (62.7%), and fourteen of the control pugs (36.8%) were males, as expected given the higher prevalence of PDM in males. The mean onset of clinical signs for the affected pugs was 78 months (standard deviation [SD] = 23), and the mean age at phenotypic confirmation of the control pugs was 132 months (SD = 19). An LD-pruned subset of the imputed variants was used (*n* = 266,687) for the Bayesian analysis, while the complete set of imputed variants (*n* = 2,140,239) was used for the XP-EHH analysis.

### 3.1. Bayesian Analysis Defined Nineteen Genetic Loci

From the 50 (0.02%) top variant effects in the BayesR model, we identified 19 associated loci, defined by variants in LD r^2^ > 0.8 with the top variant, harboring, in total, 67 genes (Table 1 and Figure 1). The associated loci with the highest variant effects and with more than one effect variant per region were chr4:11 Mb (intragenic in BicC Family RNA Binding Protein 1 *[BICC1]*), chr12:65 Mb (intragenic in SEC63 Homolog, Protein Translocation Regulator *[SEC63]*), chr17:53 Mb (top effect variant intragenic in Calsequestrin 2 *[CASQ2]*), and chr33:31 Mb (intragenic in Large 60S Subunit Nuclear Export GTPase 1 *[LSG1]*).

When comparing the risk index, i.e., the sum of the risk alleles from the 19 associated loci, between cases and controls, we found a statistically significant difference (*p* = 3.7e−12; Figure 2), with the risk index explaining 45.6% of the PDM variance (*p* = 4.3e−16;) and sex explaining 15.3% (*p* = 1.1e−07).

### 3.2. Selection Signature Analysis Defined Three Regions

The XP-EHH analysis identified three candidate regions under selection. The most significant region (*p* = 2.3e−7) was located on chromosome 15, with the top two selection variants mapping to positions chr15:28,234,106 and chr15:28,234,130. This signal included 1635 selection variants (i.e., variants with −log10(p) XP-EHH > 5) encompassing ~1.2 Mb (chr15:27,882,677–29,065,882) and overlapping the gene MGAT4 family member C *(MGAT4C)*. The XP-EHH value was negative (−5.18) and the integrated haplotype homozygosity (iHH) for allele A was higher in controls (iHH = 7.3 Mb) than in cases (iHH = 2.0 Mb), thus indicating selection in the controls. The second candidate region under selection was detected on chromosome 17. The top selection variant mapped to position chr17:57,069,701, and the signal included 19 selection variants, encompassing ~7 Kb (chr17:57,064,222–57,071,120). No genes have been annotated within this region, however, the closest genes were *SEC22* Homolog B, Vesicle Trafficking Protein *(SEC22B)* and NOTCH receptor 2 *(NOTCH2),* located 23 Kb and 44 Kb from the top selection variant, respectively. The third candidate region under selection was located on chromosome 30, with the top variant mapping to position chr30:8,951,408. This region harbored 79 selection variants, encompassing ~20 Kb (chr30:8,933,970–8,954,088), which overlapped the gene EH domain containing 4 *(EHD4)*. Both the regions on chromosome 17 and chromosome 30 indicated signals of selection in the cases (Figure 3).

## 4. Discussion

This is the first study to identify genetic loci associated with PDM, a disorder so far only reported in pugs. The BayesR analysis identified 19 associated loci harboring 67 genes, out of which more than half (*n* = 34) were putative candidate genes for PDM (Table 1). Of these 34 genes, 14 had implicated functions in more than one of the below defined categories, with possible connections to PDM pathogenesis. The XP-EHH analysis identified three candidate regions under selection, defining four candidate genes in total, within or next to the selection signals, with potential functions related to PDM. The candidate genes, potentially associated with the PDM phenotype, are implicated in bone homeostasis, cartilage, fibrosis, and inflammation.

The stability and structure of the spine is maintained by the vertebrae (bone), the vertebral endplates, the facet joints, and the intervertebral discs (cartilage). Bone is a multifunctional, dynamic, mineralized connective tissue undergoing considerable change over time. It exhibits different types of cells, including immature osteoblasts, that differentiate into long-lived osteocytes, and bone-marrow derived osteoclasts [80,81]. Osteocytes are the most common cellular components of bone and are essential for bone mass regulation. The potential loss of the structural vertebral column integrity through spinal instability has been implicated as the cause for the clinical signs of PDM. Vertebral anomalies are suggested to interfere with spinal biomechanics and to result in instability and repeated spinal cord injury [3,6,82,83,84]. Pugs have been identified to possess the progressive loss of vertebral column integrity and an increase in spinal kyphosis (the abnormal curvature of the spine) [85,86]. From the BayesR analyses, we found 24 candidate genes (50.7% of the total number of genes) with potential functions in bone homeostasis. Nine of these have previously been reported to be involved in osteoblast metabolism. For instance, Tripartite Motif Containing 38 *(TRIM38)* and TGFB Induced Factor Homeobox 1 *(TGIF1)* play critical roles in bone remodeling and are involved in osteoblast and osteoclast differentiation [41,76]. Furthermore, *BICC1,* Nescient Helix-Loop-Helix 2 *(NHLH2)*, and Protein Kinase AMP-Activated Non-Catalytic Subunit β 1 *(PRKAB1)* have been shown to affect bone mineral density [31,57,63]. In addition to being a tumor suppressor gene and being involved in inflammation, Programmed Cell Death 4 *(PDCD4)* is evolutionarily highly conserved and has a role in regulating osteogenic differentiation and bone defect repair [68].

Skeletal development, maintenance, and remodeling is regulated by osteoclasts [87]. Of the genes identified in our study, ten (14.9%) were specifically associated with the functions in osteoclasts, including *TRIM38* and *TGIF1*, discussed above. Ras Homolog Family Member U *(RhoU)* is an encoding member of the Rho family, a family of small GTPases proteins active in the organization of the actin cytoskeleton [88] and involved in migration, epithelial cell morphogenesis, and osteoclastogenesis [39,89,90,91,92]. *RhoU* has been identified in the differentiation of macrophages into osteoclasts (osteoclastogenesis), and in decreasing bone resorption by its role in the adhesion structures of osteoclasts [89]. In addition, the disruption of Mitochondrial transcription factor A *(TFAM)* has been implicated in increased bone resorption through its presence in osteoclasts [32]. Of the candidate genes from the XP-EHH analysis, *NOTCH2* is of particular interest, as it is implicated in osteoclastogenesis [93]. *NOTCH2* expression in osteoblasts has been observed to regulate the cancellous bone volume and microarchitecture [94].

Cartilage is a tissue comprised of extracellular matrix components, including compact collagen, which is a fibrous protein that adds to tissue strength and structure. It is present in the vertebral column in the intervertebral discs and vertebral endplates, and in the articular surfaces of bone [95]. Soft cartilage is a desirable characteristic in pugs; for instance, kennel clubs (e.g., the American Kennel Club and United Kennel Club; [96,97]) define the ear characteristics of the standard pug breed as thin, small, and soft, similar to black velvet. However, the selection for this trait may have affected the supportive structure in more parts of the body, including the cartilage in the spine, thereby increasing the risk for PDM. The candidate region under selection on chromosome 30, with suggested selection pressure acting in the cases, includes the gene *EHD4*, which has been implicated in the cartilage functions [98]. Fourteen (20.9%) candidate genes from the associated loci have been involved in cartilage-related processes. Six of them are directly associated with cartilage, and ten through their association with osteoarthritis, which can be described as the degradation of cartilage in the joints at the end of bones. *TGIF1*, in addition to its role in bone remodeling described above, has been directly implicated in controlling cartilage as it encodes a transcription regulator described to inhibit differentiation into cartilage [43]. *TGIF1* has also been described as playing a role in directing the differentiation of mesodermal cells (during TGFβ signaling) toward fibrogenesis instead of following chondrogenic differentiation [99]. From the XP-EHH analysis, *NOTCH2* has been observed to mediate chondrogenesis differentiation in cartilage progenitor/stem cells [100] and to play a role in chondrocyte maturation [93], and *MGAT4C* has presented with upregulated mRNA expression in osteoarthritis [36,101]. Interestingly, the XP-EHH values of the selection variants overlapping *MGAT4C* were negative, i.e., indicating selection in the controls, which may suggest that breeders are aware of the disorder and are actively breeding against it.

Seven (10.4%) of our identified candidate genes have previously been associated with the fibroblast functions, which can be linked to the excessive scar tissue of the meninges, observed in PDM. One of these genes is *TFAM*, which shows that a reduction in protein expression, and the associated mitochondrial damage, translates into an enhanced sensitivity of fibroblasts to profibrotic stimuli [38]. Fibrosis, or scarring, is a form of tissue repair in which connective tissue replaces the original parenchyma. When this process is disturbed, as in chronic diseases, it often leads to increased fibrosis [102]. The meninges are the fibrous coverings of the CNS and consist of a variety of cell types, including CNS fibroblasts and specialized immune cells. Fibroblasts from the meninges are major drivers of fibrotic scar formation following CNS injury [103].

Nine (13.4%) candidate genes from the associated loci have previously been associated with inflammatory responses, the activation of a highly coordinated immunological response specific for the initial stimulus [104]. This represents a clear link to the CNS inflammation observed in PDM. *PDCD4*, already described for its role in osteogenic differentiation above, is widely expressed in the immune cells of humans and other animals. It plays an important role in the macrophage function and exhibits both inflammatory and anti-inflammatory functions [105]. *PDCD4*-deficient mice, immunized with myelin oligodendrocyte glycoprotein to induce experimental autoimmune encephalomyelitis, have shown resistance to autoimmune encephalomyelitis and developed a reduced degree of spinal cord inflammation [106]. Interestingly, *PDCD4* promotes chronic inflammation and could therefore be relevant for the gradual and protracted onset of PDM. Furthermore, *SEC22B*, identified in the XP-EHH analysis, has been implicated in phagosome formation, crucial for the defense against pathogens [107]. In addition to being implicated in fibrosis and bone resorption, *TFAM* has also been shown to induce pro-inflammatory and cytotoxic responses of microglia [40].

We identified 19 associated loci in BayesR, defined by the 50 top effect variants; a cutoff previously used in dogs [108]. While the ratio between false negatives and false positives has been shown to be favorable for BayesR, with lower effect sizes, the risk of false positives increases [9]. A stricter definition, using the top 10 effect variants, would result in seven associated loci and 26 candidate genes, out of which 13 have potential implications in PDM and covering all of the above-mentioned biological functions. Being aware of the higher risk of false positives among the 19 associated loci, we still defined the definition of the top 50 effect variants to avoid losing the true positive associations. Given the large proportion of PDM variance explained (45.6%) by the risk index, the associated loci are indicated to be of high relevance to the development of PDM. These results will hopefully help us understand the etiology of the disease in dogs better. While there is no known human disorder corresponding to PDM, the rare disorder adhesive arachnoiditis in human [109,110] has shown similarities with PDM in terms of meningeal fibrosis and subsequent spinal cord destruction. However, no comparative studies have yet been performed. Signals on chromosome 17 were identified both in the BayesR analysis and the XP-EHH analysis. They were, however, approximately 4 Mb apart and neither the LD region nor the candidate region of selection overlapped with the other, pointing to them being different signals. Future studies should include larger sample sets to confirm our current findings and validate the lowest effect loci in particular. Analyses of the gene and protein expression differences in the relevant tissues from pugs would be of high interest to explore the functions of the candidate genes in PDM.

## 5. Conclusions

Taken together, this study suggests that the genes implicated in bone homeostasis, fibrotic scar tissue, inflammatory responses, and formation, regulation, and differentiation of cartilage are likely to be implicated in the underlying etiology of PDM. Even though the details on how the candidate genes may be involved in these biological processes remain to be fully elucidated, we believe that by utilizing the advanced genome-wide mapping methods, we have increased the knowledge about this devastating disorder.

## Figures and Tables

**Figure 1 genes-14-00385-f001:**
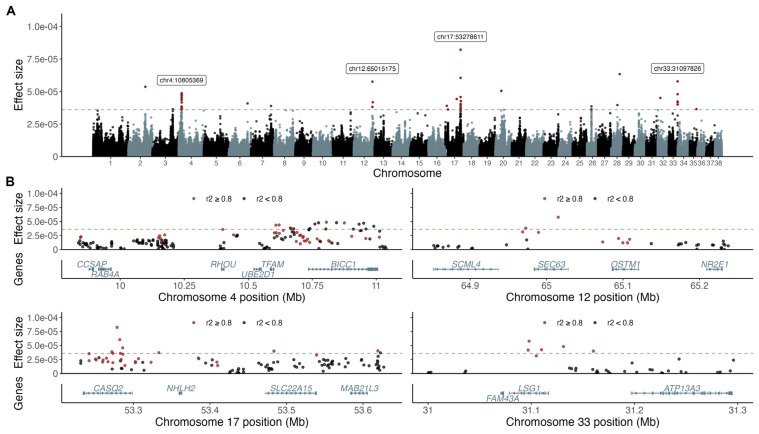
BayesR analysis resulted in four top associated loci and 19 loci in total. The top 50 effect variants are above the dotted red line defining multiple associated loci and variants with the highest effect sizes marked out with labels (**A**). Variants in LD (r^2^ > 0.8; red) with the top effect variant and annotated protein coding genes are presented for each of the four top associated loci (**B**).

**Figure 2 genes-14-00385-f002:**
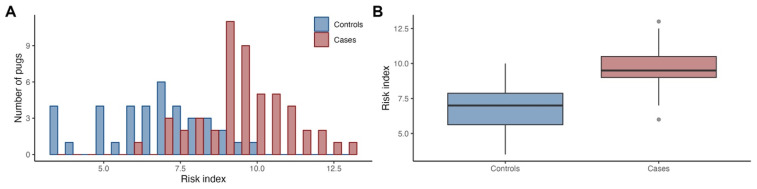
The risk index was higher in PDM cases compared to controls. The histogram shows the distribution of summed risk alleles at the associated loci (risk index) in cases and controls (**A**), and the boxplots indicate the median, first and third quartiles, and range (**B**).

**Figure 3 genes-14-00385-f003:**
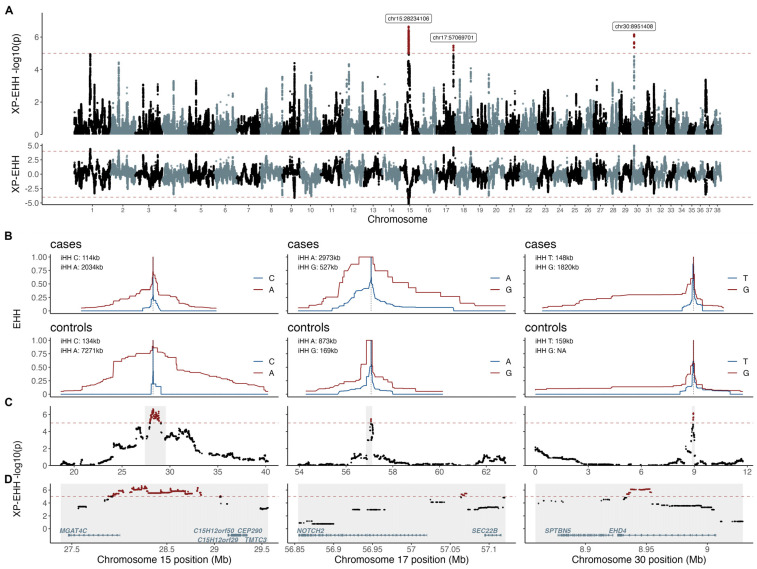
XP-EHH analysis defined three candidate regions under selection. XP-EHH *p*-values and XP-EHH values are presented across the whole genome (**A**). The candidate regions under selection are presented as EHH plots in cases and controls for chromosomes 15, 17, and 30 with haplotype lengths indicated by integrated extended haplotype homozygosity (iHH) for the respective alleles in cases and controls (**B)**. XP-EHH *p*-values corresponding to the iHH regions (**C**) and zoom-ins around the top selection variants (shaded in grey) are presented with the annotated protein coded genes in (**D**).

**Table 1 genes-14-00385-t001:** PDM associated loci identified with BayesR.

Top Variant	A1/A2	Effect Size ^1^	EAF A/U	VAR EXP (%)	Variants in Top 50 (N)	LD Region ^2^ (Size in Kb)	Genes Closest to Top Variant	Genes in LD Region	Bone Homeostasis	Cartilage	Osteoarthritis	Fibrotic Scar Tissue	Inflammatory Response
chr2:60108854	G/A	5.36E−05	0.42/0.59	4.77	1	chr2:60095511–60653394 (558)	*CES5A* (intergenic 55 Kb)*; GNAO1* (intergenic 128 Kb)	*CAPNS2, CES1, CES5A, IRX6, LPCAT2, MMP2, SLC6A2*	*CES1* [25], *MMP2* [26] [27], *IRX6* [28]		*GNAO1* [29]		
chr3:71553918	A/G	−3.64E−05	0.58/0.34	13.57	1	chr3:71521315–71768914 (248)	*APBB2*	*APBB2*			*APBB2* [30]		
chr4:10805369	C/T	4.88E−05	0.14/0.37	10.83	17	chr4:9844757–11012379 (1168)	*BICC1*	*BICC1, CCSAP, RAB4A, RHOU, TFAM, UBE2D1*	*BICC1* [31], *TFAM* [32], *RHOU* [33] [34]		*TFAM* [35] [36], *BICC1* [37]	*TFAM* [38], *RHOU* [39]	*TFAM* [40]
chr6:65661296	T/C	−4.09E−05	0.16/0.04	4.67	1	chr6:65386176–65714252 (328)	*ADGRL2*	*ADGRL2*					
chr7:70300935	A/G	−3.90E−05	0.22/0.09	1.66	1	chr7:69701081–71152660 (1452)	*DLGAP1*	*DLGAP1, MYL12B, MYOM1, TGIF1*	*TGIF1* [41], *MYL12B* [42]	*TGIF1* [43]	*MYOM1* [37]		*MYL12B* [44]
chr12:65015175	C/T	5.76E−05	0.21/0.41	3.80	2	chr12:64968699–65110915 (142)	*SEC63*	*SEC63, OSTM1*	*OSTM1* [45]				
chr12:66289122	G/A	4.18E−05	0.16/0.38	0.27	1	chr12:66266562–66343853 (77)	*CD164* (intergenic 6 Kb)*; CCDC162P* (intergenic 5 Kb)	*CCDC162P/C12H6orf183, CD164, PPIL6*	*CD164* [46]	*CD164* [46]	*CD164* [47]		*CCDC162P* [48]
chr17:5061849	A/G	3.90E−05	0.29/0.46	3.67	1	chr17:4062154–5436291 (1374)	*RNF144A* (intergenic 424 Kb)*; ID2* (intergenic 895 Kb)	*CMPK2, RNF144A, RSAD2*	*ID2* [49]	*RNF144A* (as *GRASLND*, i.e., *RNF144-AS1*; human transcript) [50]			*RSAD2* [[51],[52] and [53]]
chr17:9192869	G/C	−3.62E−05	0.33/0.12	7.78	1	chr17:8871659–9520837 (649)	*TRIB2* (intergenic 33 Kb)	*TRIB2*		*TRIB2* [54]			
chr17:39587560	A/G	−4.43E−05	0.19/0.07	1.64	1	chr17:39502199–39589843 (88)	*SH2D6* (intergenic 15K Kb), *MAT2A*(intergenic 55 Kb)	*GGCX, MAT2A*	*GGCX* [55]				
chr17:53278611	T/A	8.22E−05	0.28/0.53	1.44	11	chr17:53239890–53619454 (380)	*CASQ2, SLC22A15*	*CASQ2, MAB21L3, NHLH2, SLC22A15*	*CASQ2* [56], *NHLH2* [57]	*CASQ2* [58]	*CASQ2* [59]		
chr17:54018787	C/T	−4.27E−05	0.54/0.30	0.06	1	chr17:54018787–54024163 (5)	*IGSF3*	*IGSF3*					*IGSF3* [60]
chr20:22144318	A/G	−5.05E−05	0.42/0.27	0.31	1	chr20:21579384–22373451 (794)	*MITF* (intergenic 271 Kb), *FRMD48* (intergenic 258 Kb)	*FRMD4B, MITF*	*MITF* [61]			*MITF* [62]	
chr26:15777160	C/T	3.86E−05	0.01/0.14	5.00	2	chr26:15350328–16204269 (854)	*CIT*	*BICDL1, CCDC60, CIT, GCN1, PRKAB1, PXN, RAB35, RPLP0, TMEM233*	*PKAB1* [63], *PXN* [64], *TMEM233* [65]		*PXN* [66]	*PXN* [67]	
chr28:21986613	C/T	3.96E−05	0.44/0.57	0.03	1	chr28:21986613–22282079 (295)	*RBM20*	*BBIP1, PDCD4, RBM20, SHOC2*	*PDCD4* [68]		*PCDC4* [69]	PDCD4 [70], SHOC2 [71]	*PDCD4* [72]
chr28:30266337	G/T	−6.34E−05	0.33/0.24	0.47	1	chr28:30041108–30695769 (655)	*ARF1* (intergenic 26 Kb), *PLPP4* (intergenic 244 Kb)	*SEC23IP, PLPP4*	*SEC23IP* [73], *ARF1* [74]				
chr32:9378372	C/T	−4.51E−05	0.28/0.22	0.41	1	chr32:9192329–9505692 (313)	*ARHGAP24*	*ARHGAP24*					*ARHGAP24* [75]
chr33:31097826	C/T	−5.78E−05	0.10/0.04	0.63	5	chr33:31088620–31160245 (72)	*LSG1*	*LSG1*					
chr35:23618123	T/C	−3.65E−05	0.44/0.41	0.02	1	chr35:23525268–23956076 (431)	*SCGN*	*CARMIL1, HIST1H2AA, HIST1H2BA, SCGN, SLC17A1, SLC17A2, SLC17A3, SLC17A4, TRIM38*	*TRIM38* [76], *SLC17A2* (i.e., *SLC34A1*) and *SLC17A1* [77]	*CARMIL1* [78]		*CARMIL1* [78]	*CARMIL1* [78], *TRIM38* [79]

^1^ Effect size off A1; ^2^ LD r^2^ > 0.8; Abbreviations: EAF effect allele frequency; VAR EXP variance explained.

## Data Availability

The genotype datasets (plink files: bed, bim, fam, pheno), after quality control and relatedness filtering, have been uploaded onto the SciLifeLab data repository (DOI 10.17044/scilifelab.21948521).

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
