# Peer review of "Multiple Genetic Loci Associated with Pug Dog Thoracolumbar Myelopathy"

_genes, 2023, doi:10.3390/genes14020385_

Round 1

Reviewer 1 Report

Very good study. The materials and methods are repeatable and the contribution is clear. Approximately 43 % of citations are published recently within five years. And around 87 % are from the last fifteen years. Considering the topic, I think that it is very sufficient amount of recent references.

For Table 1 A it would be clearer to indicate an arrow to a specific SNP in the graph.

Species name should be italicized in the references section – specifically reference No. 72 “Celastrus orbiculatus”, line 527.

Author Response

Response to Reviewer 1 Comments

Very good study. The materials and methods are repeatable and the contribution is clear. Approximately 43 % of citations are published recently within five years. And around 87 % are from the last fifteen years. Considering the topic, I think that it is very sufficient amount of recent references.

Response: We thank the reviewer for the very encouraging remarks.

For Table 1 A it would be clearer to indicate an arrow to a specific SNP in the graph.

Response: We tried adding an arrow pointing to the individual SNPs in the graph, however, we found that it risked muddling the position of the SNP itself. Instead, we positioned the label centred above the individual SNPs. We did this also for Figure 3. We hope that this sufficiently makes the figures easier to interpret.

Species name should be italicized in the references section – specifically reference No. 72 “Celastrus orbiculatus”, line 527.

Response: Thank you for noticing this. We have formatted the species name in reference number 72 to italic.

Reviewer 2 Report

The study conducted by Brander et al. has aimed to identify genes associated with thoracolumbar myelopathy (PDM) in pug dogs. A genome-wide analysis based on Bayesian model for mapping complex traits and a cross-population extended haplotype homozygosity test was applied. Control and studied groups were rather small – 38 controls and 51 cases. The topic is interesting, since there is no information about the genetic background of this disorder. The Authors identified 19 loci and 3 candidate regions under selection. These loci harbor more than 30 genes which can be candidate genes for PDM. The study is well designed, results are convincing.

Minor remarks:

-          Some additional information in Introduction about the prevalence of PDM in pugs  and what is known about the genetic background of this disorder in humans would be useful.

-          Description of study populations should be included in M&M, it is not clear why only limited data (sex, age etc) are presented for final populations in result section

-          the table 1 is hard to read - should be improved

-          why X chromosome was not included in the analysis? It is important if the disorder is more common in males. Cases of hereditary X-linked thoracic myelopathy were reported in human.

-          The most interesting candidate region seems to be located on chromosome 17, since was indicated by both approaches (BayesR, XP-EHH) - requires more detailed discussion

-          Discussion part is poorly written – e.g.  all paragraphs start the same -  “Bone is… a tissue”,  “Osteoclasts function ..”, “Cartilage is a tissue”, “Fibrosis ….is a form of tissue”, “Inflammation involves” - needs improvement. 

Author Response

Response to Reviewer 2 Comments

The study conducted by Brander et al. has aimed to identify genes associated with thoracolumbar myelopathy (PDM) in pug dogs. A genome-wide analysis based on Bayesian model for mapping complex traits and a cross-population extended haplotype homozygosity test was applied. Control and studied groups were rather small – 38 controls and 51 cases. The topic is interesting, since there is no information about the genetic background of this disorder. The Authors identified 19 loci and 3 candidate regions under selection. These loci harbor more than 30 genes which can be candidate genes for PDM. The study is well designed, results are convincing.

Response: We would like to thank the reviewer for the summary and the reassuring comments.

Minor remarks:

-          Some additional information in Introduction about the prevalence of PDM in pugs and what is known about the genetic background of this disorder in humans would be useful.

Response: There is no currently known prevalence of the disorder, other than clinical [unpublished] observations from veterinarians for the disorder to be fairly common. We have addressed this in the introduction (p1, lines 43-45). Likewise, there is no certain equivalent disorder in humans, although there are similarities with the rare human disorder adhesive arachnoiditis. This has been added to the discussion (p10, lines 430-435).

-          Description of study populations should be included in M&M, it is not clear why only limited data (sex, age etc) are presented for final populations in result section

Response: We have added information about the study population, prior to quality control, in the M&M section (p3, lines 104-107).

-          the table 1 is hard to read - should be improved

Response: Thank you for pointing this out! By spending some extra time with it, we have condensed some information and reformatted a lot in order to make it more eligible. We hope that the reviewer agrees with us that it is now a lot easier to take in.

-          why X chromosome was not included in the analysis? It is important if the disorder is more common in males. Cases of hereditary X-linked thoracic myelopathy were reported in human.

Response: The question regarding the omission of the X chromosome in the analysis is as valid and important question as it is complex. Due to the imbalance of chromosome X in males and females, there is a risk of introducing bias into the association analysis, and for this reason it has historically been excluded in many GWASs (Tukiainen et al., 2014; Wise, A. L., Gyi, L., & Manolio, T. A., 2013).

In order to balance the allele dosages between the sexes, there is a random X chromosome inactivation, silencing one of the two chromosomes in females. At the same time, the silencing is not complete, but about 15% remains unsilenced, complicating adjusting for silencing. Furthermore, the X chromosome includes non-pseudoautosomal (NPAR) and pseudoautosomal (PAR) regions. Therefore, the X chromosome ideally needs to be split into PAR and NPAR regions and analyzed separately as i) an autosomal chromosome and ii) X chromosome, respectively. This is particularly relevant for the quality control steps prior to association analysis, which have the purpose to remove low-quality genetic variants and genotypes. For instance, in the NPAR, males are hemizygous, which causes the intensity of allele signals from SNPChips or the depth of coverage from resequencing to be half of that of females, thus leading to a skewed proportion of missing variants between the two sexes. Similar issues can be predicted also when applying filters for Hardy-Weinberg Equilibrium and minor allele frequencies, because of the different expected frequencies in the two sexes.

To conclude, when including the X chromosome in association analyses this all needs to be accounted for, which is not technically straightforward and easily interpretable. For these reasons, as well as to eliminate the risk of obtaining biased results, we did not set out to address this within this study. However, exploring the role of the X chromosome would certainly be of great interest for this complex disease and should be considered for future follow-up studies.

-          The most interesting candidate region seems to be located on chromosome 17, since was indicated by both approaches (BayesR, XP-EHH) - requires more detailed discussion

Response: Thank you for pointing out this omission! The signals are in chromosome 17 are approximately 4 Mb apart with no overlap between neither the LD region from the BayesR analysis nor the candidate region of selection from the XP-EHH analysis. We therefore have to conclude that it is the matter of two different signals. We have now addressed this in the discussion (p10, lines 435-438).

-          Discussion part is poorly written – e.g.  all paragraphs start the same -  “Bone is… a tissue”,  “Osteoclasts function ..”, “Cartilage is a tissue”, “Fibrosis ….is a form of tissue”, “Inflammation involves” - needs improvement. 

Response: We would like to thank the reviewer for pointing this out. This is a discussion dense with descriptions and information. We have tried our best to improve the flow and reduce the repetitiveness of the text.

References

Tukiainen, T., Pirinen, M., Sarin, A. P., Ladenvall, C., Kettunen, J., Lehtimäki, T., Lokki, M. L., Perola, M., Sinisalo, J., Vlachopoulou, E., Eriksson, J. G., Groop, L., Jula, A., Järvelin, M. R., Raitakari, O. T., Salomaa, V., & Ripatti, S. (2014). Chromosome X-Wide Association Study Identifies Loci for Fasting Insulin and Height and Evidence for Incomplete Dosage Compensation. PLoS Genetics, 10(2). https://doi.org/10.1371/journal.pgen.1004127

Wise, A. L., Gyi, L., & Manolio, T. A. (2013). EXclusion: Toward integrating the X chromosome in genome-wide association analyses. In American Journal of Human Genetics (Vol. 92, Issue 5). https://doi.org/10.1016/j.ajhg.2013.03.017
